# The Effect of Body Composition on Gait Variability Varies with Age: Interaction by Hierarchical Moderated Regression Analysis

**DOI:** 10.3390/ijerph19031171

**Published:** 2022-01-21

**Authors:** Yungon Lee, Sunghoon Shin

**Affiliations:** 1Research Institute of Human Ecology, Yeungnam University, Gyeongsan-si 38541, Korea; lyg2311@ynu.ac.kr; 2Neuromuscular Control Laboratory, Yeungnam University, Gyeongsan-si 38541, Korea; 3School of Kinesiology, College of Human Ecology & Kinesiology, Yeungnam University, 221ho, 280 Daehak-ro, Gyeongsan-si 38541, Korea

**Keywords:** body composition, gait variability, age, interaction

## Abstract

Although body composition has been found to affect various motor functions (e.g., locomotion and balance), there is limited information on the effect of the interaction between body composition and age on gait variability. The purpose of this study was to determine the effect of body composition on gait according to age. A total of 80 men (40 young and 40 older males) participated in the experiment. Body composition was measured using bioelectrical impedance analysis (BIA), and gait parameters were measured with seven-dimensional inertial measurement unit (IMU) sensors as each participant walked for 6 min at their preferred pace. Hierarchical moderated regression analysis, including height as a control variable and age as a moderator variable, was performed to determine whether body composition could predict gait parameters. In young males, stride length decreased as body fat percentage (BFP) increased (R^2^ = 13.4%), and in older males, stride length decreased more markedly as BFP increased (R^2^ = 26.3%). However, the stride length coefficient of variation (CV) of the older males increased significantly as BFP increased (R^2^ = 16.2%), but the stride length CV of young males did not change even when BFP increased. The increase in BFP was a factor that simultaneously caused a decrease in gait performance and an increase in gait instability in older males. Therefore, BFP is more important for a stable gait in older males.

## 1. Introduction

Human locomotion is a complex process caused by interactions of the neuromuscular system [1]. Human gait is affected by changes in the central nervous system (CNS), sensory-motor system control, or body composition [2,3,4,5]. In general, evidence from basic, clinical, and epidemiological studies points to aging-induced CNS degradation as an important contributor to mobility limitation in older persons without overt neurological disease [3]. In addition, sensory signals in the sensory-motor system compensate for the deteriorated walking performance of older persons by strengthening the transmission of gait commands [6]. In contrast, changes in body composition, such as body fat and muscle mass, deteriorate gait performance [7,8,9,10]. For example, compared to healthy controls, young adults with a higher body mass index (BMI) showed an increase in stance phase duration (2.9%) and double support time (18.2%) [11], and middle-aged men with high body fat showed a decrease in stride length (9.5%) [12]. In addition, the stride length while walking decreased in older persons with low leg muscle mass [8], and as the leg muscle mass decreased, the maximum heel clearance (40%) in young adults decreased [10]. 

In general, changes in gait with neurological aging, which exist in older persons without specific neurological diseases, are considered part of a natural aging process [13]. For example, healthy older individuals generally walk with a shorter step length and wider step width than healthy adults [14]. In healthy older individuals, body fat is a potential physical factor that can change the gait mechanism. For example, a previous study reported that the higher the BMI, the lower the joint power at the ankle in the anterior-posterior (AP) direction and the higher the joint power at the hip in the medio-lateral (ML) direction at the preferred walking speed [15]. As an increased force in the AP direction in gait mainly induces an efficient gait, an unnecessary increase in joint power in the ML direction reflects an inefficient gait in obese older persons. In particular, it has been shown that direct joint burden due to increased load can lead to neuromuscular degeneration of the lower extremities (i.e., knee arthritis) in older men [16]. Therefore, given the effects of body composition on gait and the interaction between aging and body composition, the effects of body composition on gait are likely to be even more severe in older persons, who are more sensitive to changes in body composition.

Recently, gait variability has been reported as a relatively stronger predictor of neuromuscular system deterioration or aging than walking performance [17]. Gait variability, known as fluctuations in human movement [18], can be quantified as the standard deviation (SD) or the coefficient of variation (CV) of spatiotemporal gait parameters [19,20]. In general, gait variability increases with age [21]. For example, the step time SD and step length SD linearly increased with age for the older persons between 60 and 86 years of age, regardless of height, weight, and presence of chronic disease [22]. In addition, older persons showed increased stride width SD compared to young adults regardless of walking speed (slow, normal, and fast) [23]. As age-induced gait variability is closely related to mobility restriction or fall risk [24,25,26], finding the factors contributing to gait variability in advance can reduce mobility restriction or fall risk during gait [27]. Therefore, gait variability is an important indicator that must be considered when evaluating the interaction between body composition and age.

It seems that human gait can be considerably changed by the interaction of body composition and age, but the interaction between these two factors was not clear in previous studies. In particular, it is unclear whether gait variability increases or decreases because of the interaction between body composition and age. Therefore, the purpose of this study was to determine the effect of body composition on gait according to age. To confirm the effect of age more clearly, the effect of gender must be excluded. A previous study found that height-controlled gait speed and stride length did not differ solely by gender in young and older adults [28,29]. As a result, this study’s gender was limited to men. Finally, we hypothesized that the effect of body composition on gait varies with age and is expected to have a significant impact on gait variability in older males.

## 2. Materials and Methods

### 2.1. Participants

This study was designed as a cross-sectional study. A total of 80 male subjects participated in the gait experiment and were divided into a young male group (*n* = 40) and an older male group (*n* = 40) (Figure 1). Participants were selected from a pool of healthy male adults living in the community. The recruitment method used was a community notice. Participants were chosen based on their ability to walk independently and their willingness to engage in physical activity. Those with a neuromuscular disease who had an artificial joint or metal device inserted were excluded. A random sampling method was used to select participants. Young men ranged in age from 20 to 30 years [30], while older men ranged in age from 55 to 85 years [31,32,33,34,35]. Due to time constraints, two young men who did not receive their assigned intervention were dropped out. One older man was unable to participate due to back pain. In addition, one older man dropped out of the experiment due to dizziness while walking, but no side effects were reported by any of the subjects after the study was completed. Their specific physical characteristics are presented in Table 1. This study was conducted in a university indoor gymnasium. After their body composition was evaluated, all of the participants participated in the gait experiment. The participants signed an informed consent form prior to participating in the experiment. This study was approved by a Bioethics Committee (IRB-2018-09-003-002). 

### 2.2. Body Composition

Body composition was measured using an InBody 520 (Biospace Co., Ltd., Seoul, Korea). The InBody 520 estimates the body composition of the human body based on bioelectrical impedance analysis (BIA). The bioelectrical resistance method is useful in body composition research because the electrode in contact with the human body measures the resistance value (impedance) of the body through electrical current [36]. It, therefore, allows for easy and non-invasive measurement of body composition [37,38]. The participants climbed on the InBody, faced the front, and stood in an upright position for approximately 60 s. Finally, the body mass index (BMI), body fat percentage (BFP), and skeletal muscle mass (SMM) were determined from the results of the body composition analysis.

### 2.3. Gait

Gait was measured using a seven-dimensional (three-dimensional accelerometer + three-dimensional gyroscope + one-dimensional barometer) inertial measurement unit (IMU) sensors (Physilog5®, GaitUp^™^, Lausanne, Switzerland). The study participants walked a straight course in the gym at their preferred speed for 6 min with 7-axis IMU sensors attached to the instep of both feet in a vertical direction. After 6 min, gait kinematic data were collected from the IMU sensor, and average gait speed, stride time, and stride length were obtained. Furthermore, gait variability parameters, which were indexed by the coefficient of variation (CV) of gait variables obtained, were calculated as (standard deviation/mean) × 100.

### 2.4. Statistical Analysis

First, the Kolmogorov–Smirnov test of normality was performed, and the physical characteristics, body composition, and gait of the young and older male groups were compared using the independent sample *t*-test and Mann–Whitney test. The reason for conducting the *t*-test and Mann-Whitney test in advance was to extract the potential influencing factors on the gait. VIF (variance inflation factor) was accessed for independent variables. All data were converted to Z-values (normalized) before being entered into the regression model. Hierarchical moderated regression analysis was performed to determine whether body composition can predict the performance and variability of spatiotemporal gait variables. In this case, height and age were considered as the control variable and moderator variable, respectively. First, in Model 1 of the hierarchical moderated regression analysis, the independent variables (BFP and SMM) and the control variable (height) were input. Subsequently, in Model 2, the moderator variable (age) was included. Then, in Model 3, the interaction variables (BFP and SMM × age) were added. Finally, if there was an interaction effect (i.e., a significant moderating effect) from Model 3, the young and older male groups were separated, and each linear regression analysis was performed and presented as a graph (Figure 2). The statistical significance level was set at *p* < 0.05, and SPSS 23 (IBM Inc., Armonk, NY, USA) was used for statistical analysis. The sample size for the multiple regression analysis was calculated using G * power 3.1.9.4 software (Heinrich Heine Düsseldorf University, Düsseldorf, Germany) set to a significance level (α) = 0.05, power (1 − β) = 0.80, and medium effect size (*f*^2^) = 0.15. Therefore, the optimal sample size required was estimated to be 77 persons. In the regression model, the effect size was calculated as Cohen’s *f*^2^, and the *f*^2^ values were categorized as small (0.02–0.14), medium (0.15–0.34), and large (≥0.35) sizes [39].

## 3. Results

### 3.1. Independent Sample T-Test and Mann–Whitney Test

The results of these tests are presented in Table 1. Demographic information revealed significant differences in age, height, and weight between young and older males (*p* < 0.05). In addition, there was a significant difference in BFP and SMM between young and older males in terms of body composition (*p* < 0.01). However, there was no significant difference in gait speed, stride time, stride length, stride time CV, and stride length CV between young and older males. 

### 3.2. Hierarchical Moderated Regression Analysis

Table 2 shows the results of the hierarchical moderated regression analysis used to predict stride length. The predictor variable affecting stride length in Model 1 was BFP (R^2^ = 0.185; *p* < 0.01; *f*^2^ = 0.22), and the predictor variable affecting stride length in Model 2 was also BFP (R^2^ = 0.197; *p* < 0.01; *f*^2^ = 0.25). In Model 3, the predictors affecting stride length were BFP and the interaction variable (BFP × age) (R^2^ = 0.268; *p* < 0.01; *f*^2^ = 0.37). The results of the interaction effects are presented in Figure 2A. In the young male group, the stride length was significantly shorter as the BFP increased (R^2^ = 0.134; *p* < 0.05; *f*^2^ = 0.15), and in the older male group, the stride length was significantly smaller as the BFP increased (R^2^ = 0.263; *p* < 0.01; *f*^2^ = 0.36). However, none of the predictors of body composition affected the stride time.

Table 3 shows the results of the hierarchical moderated regression analysis for predicting the stride length CV. In Model 1, there were no predictors affecting stride length CV, but in Model 2, the predictors affecting stride length CV were BFP and SMM (R^2^ = 0.125; *p* < 0.05; *f*^2^ = 0.14). In Model 3, the predictors affecting stride length CV were BFP, SMM, and age, and the interaction variable (BFP × age) (R^2^ = 0.202; *p* < 0.01; *f*^2^ = 0.25). The results of the interaction effect are shown in Figure 2B. In the older male group, the stride length CV increased significantly as the BFP increased (R^2^ = 0.162; *p* < 0.01; *f*^2^ = 0.19), but there was no significant change in the stride length CV in the young male group. However, none of the predictors of body composition affected the stride time CV.

## 4. Discussion

This study aimed to confirm the relationship between body composition measured by the BIA and the kinematic gait variables measured by the IMU sensor during 6-min ground walking in young and older males. Specifically, the purpose of this study was to investigate how the effect of body composition on gait varies with age, which is a moderator variable. We hypothesized that the effect of body composition on gait variability would differ according to age. The results of this study can be summarized as follows.

First, there was a tendency for stride length to decrease as BFP increased, regardless of age, but the effect was significantly greater in the older males group. In the regression analysis, the explanatory power of BFP for stride length in young males was 13.4%, and the explanatory power of BFP for stride length in older males was 26.3% (Figure 2A). Some of the causes of the interaction between BFP and age have been considered in previous studies. For example, although BMI and BFP are the same for young and older adults, obesity sites tend to differ according to age [40], resulting in different gait mechanisms. Obesity in older persons mainly consists of abdominal obesity (i.e., the state of accumulation of visceral fat) [40], and abdominal obesity in the older persons is particularly detrimental to physiological changes (decreased metabolism, increased inflammation, and hormonal imbalance), resulting in senescence syndrome (frailty syndrome) or sarcopenia [41,42]. These diseases tend to cause a decline in lower extremity muscle strength [43,44], and it is possible that this obesity-induced decline in lower extremity muscle strength in older persons significantly reduces the stride length in gait. In addition, it is possible that the direct joint burden caused by weight gain in the older persons changes the biomechanical characteristics of gait. The increase in the range of rotation in the hip in the ML direction and the decrease in the peak joint moment in the ankle in the AP direction during walking in obese older persons fundamentally impede forward walking [15], resulting in a decrease in stride length. In addition, the gait strategy for preventing falls due to obesity in the two age groups may be different [45]. Although young adults adopt an active and bold gait strategy despite weight gain [10], obese older persons exhibit a passive gait pattern that significantly reduces speed, widens stride width, and slightly lengthens stance duration [15,46]. This passive gait of obese older individuals is considered an effort to secure stability from the risk of falling by reducing the stride length. These findings are similar to those of previous studies that obese older individuals exhibit reduced stride length [45,46,47,48], suggesting that gait changes due to obesity may vary according to age. Hence, an increase in BFP appears to decrease the stride length of the older males more than that of relatively young males.

Second, this study showed that the effects of BFP on gait variability in young and older males are different. The increase in BFP increased the stride length CV in the older males but did not affect the stride length CV in young males. In other words, in the regression analysis, the explanatory power of BFP for the stride length CV of young males was 0.1%, but the explanatory power of BFP for the stride length CV of the older males was 16.2% (Figure 2B). There are several possible reasons for these findings. In general, an increase in BFP in the older persons increases the body fat mass of the lower extremities while decreasing the muscle mass of the lower extremities [49]. This reduction in muscle strength is known to increase motor variability as it is associated with fewer motor units and higher firing rates [50,51]. Therefore, a decrease in lower extremity muscle strength due to an increase in BFP in the older persons may have increased the stride length CV [52]. In addition, an increase in body fat mass in older persons tends to worsen cognitive function [53]. This cognitive decline is due to decreased cognitive processing speed and attention [54,55]. In particular, a decrease in cognitive processing speed disrupts the regular pattern of gait control related to the CNS [56], and attention segmentation has been reported to significantly deteriorate accommodative ability in gait timing [57]. Although this study did not evaluate the cognitive function of the older persons, it is possible that the cognitive decline due to the increase in BFP in older persons increased the stride length CV. In addition, the physical pressure of the joint due to weight-bearing can sometimes damage the surrounding tissues [58], and this lowering of proprioception in the older persons can impact the position or movement of the lower extremities in contact with the ground during locomotion [59]. Therefore, the decline in proprioception in older persons can be considered a contributing factor to the increase in stride length CV. The findings of this study were similar to those of a previous study, in which the step length variability increased as the BMI of the older persons increased [60], suggesting that the effect of BFP on gait variability differs according to age. Hence, an increase in BFP appears to significantly increase the stride length CV only in older males.

Finally, this study found that body composition had no relationship with stride time or stride time CV. That is, increasing BFP decreased stride length, a spatial parameter of both young and old men, and increased stride length CV in older men, but it had no effect on the temporal parameters, stride time, and stride time CV. According to the findings of previous studies, regardless of age, the stride parameter has a more significant correlation with the obesity factor in the spatial variable than in the temporal variable [11,12,15,46,47,48].

## 5. Study Limitations

This study has strengths and limitations. We consider it a strength that gait was assessed during 6 min natural walking not short length walking in the lab which could be close to real walking. However, we did not measure angular kinematic and kinetic data, so there were limitations to interpreting the results from multi-joint mechanisms during gait. The subjects in this study were all males. Gender was not taken into account. It was possible to reduce the gender deviation, but there could be differences in the results by gender. Thus there is a limit to generalizing the conclusion.

## 6. Conclusions

The effect of BFP on stride length CV differed significantly by age in this study. Stride length CV increased significantly with increasing BFP in older men, but there was no change in stride length CV with increasing BFP in younger men. BFP does not determine time parameters, which are typically expressed as stride mean and CV. This means that an increase in BFP reduced young men’s walking ability slightly but had no effect on gait stability. However, in older men, this resulted in both poor gait performance and gait instability. This does not imply changes in coordinate patterns that structurally alter stride time variability, which is an important feature of locomotion. Our findings support the effect of mechanical BFP impairment on gait regularity in the older persons, as evidenced by a BFP-dependent increase in spatial gait variability in obese older persons, despite the fact that obesity does not cause fundamental changes in temporal coordination patterns. In conclusion, although it has no effect on temporal modulation during gait, managing BFP, a body composition index, is more important in older men than in younger men for stable gait.

## Figures and Tables

**Figure 1 ijerph-19-01171-f001:**
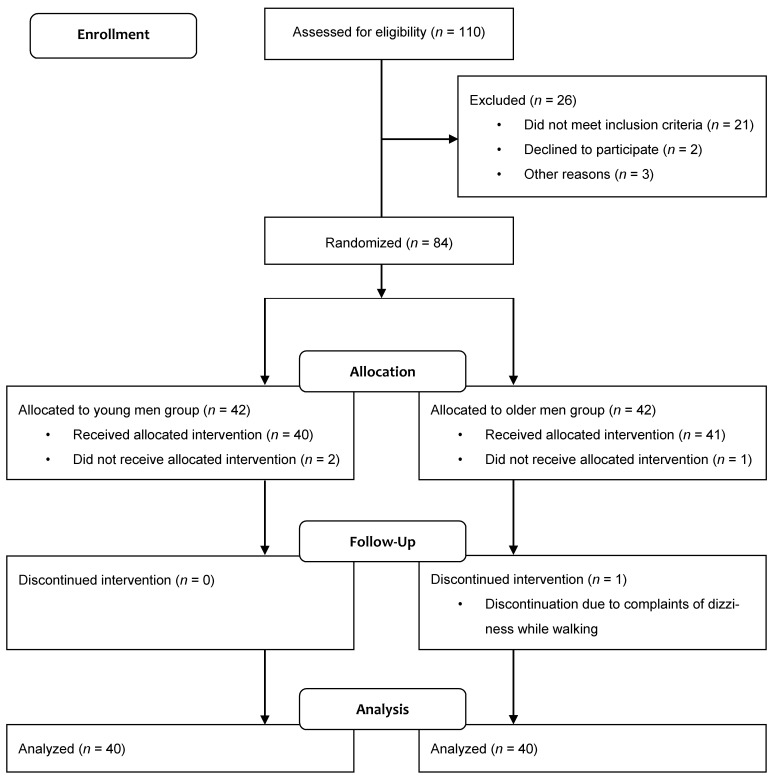
Flowchart showing the experimental design of this study.

**Figure 2 ijerph-19-01171-f002:**
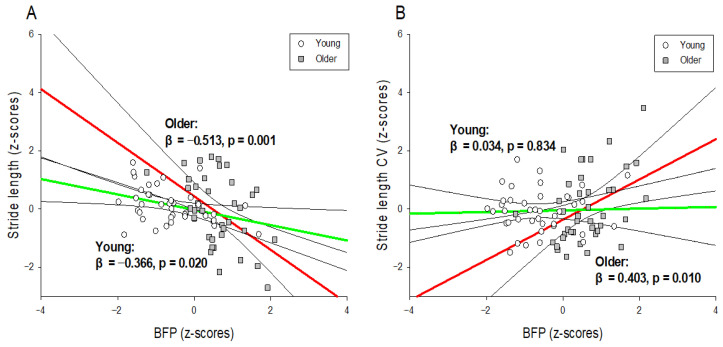
Interaction graphs for hierarchical moderated regression analysis. The association between normalized gait variables ((**A**): gait performance and (**B**): gait variability) and normalized BFP according to age (young and older males). The linear regression line for older males is red, and that for young males is green. The black line is the 95% confidence interval of the regression line. BFP: body fat percentage; CV: coefficient of variation.

**Table 1 ijerph-19-01171-t001:** Demographic information of the male participants.

Characteristics	Young (*n* = 40)	Older (*n* = 40)	Total (*n* = 80)	*p*-Value
Age (years)	22.25 ± 2.23	74.05 ± 6.86	48.15 ± 26.55	<0.001 *
Height (cm)	174.94 ± 5.05	168.35 ± 5.38	171.64 ± 6.15	<0.001 *
Weight (kg)	73.98 ± 10.53	69.23 ± 8.15	71.60 ± 9.66	0.027 *
BMI (kg/m^2^)	23.35 ± 4.08	24.55 ± 2.76	23.95 ± 3.51	NS
BFP (%)	17.06 ± 6.32	26.28 ± 5.25	21.67 ± 7.41	<0.001 *
SMM (kg)	34.73 ± 3.62	28.02 ± 3.25	31.38 ± 4.80	<0.001 *
Gait speed (m/s)	1.27 ± 0.11	1.27 ± 0.17	1.27 ± 0.15	NS
Stride time (s)	1.06 ± 0.07	1.03 ± 0.05	1.05 ± 0.06	NS
Stride length (m)	1.34 ± 0.08	1.30 ± 0.17	1.32 ± 0.13	NS
Stride time CV (%)	2.18 ± 0.49	2.37 ± 0.67	2.27 ± 0.59	NS
Stride length CV (%)	3.48 ± 0.70	3.61 ± 1.17	3.54 ± 0.96	NS

Data are mean ± SD. BMI: body mass index; BFP: body fat percentage; SMM: skeletal muscle mass. CV: coefficient of variation. * indicates significant difference (*p* < 0.05). NS: not significant. The p-values are significant differences between young and older males.

**Table 2 ijerph-19-01171-t002:** Summary of the results of the hierarchical moderated regression analysis for predicting stride length mean (m).

Predictors	Total (*n* = 80)					
	Model 1		Model 2		Model 3	
	*β*	*p*	*β*	*p*	*β*	*p*
Height (cm)	0.231	NS	0.190	NS	0.114	NS
BFP (%)	−0.397	0.001 ***	−0.485	0.001 ***	−0.563	0.001 ***
SMM (kg)	−0.222	NS	−0.088	NS	−0.093	NS
Age (years)			0.193	NS	0.216	NS
BFP × Age					−0.277	0.010 **
SMM × Age					−0.037	NS
R^2^ block 1 = 0.185	ΔR^2^ = 0.185					
R^2^ block 2 = 0.197	ΔR^2^ = 0.012					
R^2^ block 3 = 0.268	ΔR^2^ = 0.071					

BFP: body fat percentage; SMM: skeletal muscle mass. ΔR2: R-square change. ** *p* < 0.01; *** *p* < 0.001. NS: not significant.

**Table 3 ijerph-19-01171-t003:** Summary of the results of the hierarchical moderated regression analysis for predicting stride length CV (%).

Predictors	Total (*n* = 80)					
	Model 1		Model 2		Model 3	
	*β*	*p*	*β*	*p*	*β*	*p*
Height (cm)	0.149	NS	0.227	NS	0.305	NS
BFP (%)	0.171	NS	0.336	0.026 *	0.416	0.007 **
SMM (kg)	−0.284	NS	−0.536	0.022 *	−0.531	0.019 *
Age (years)			−0.363	NS	−0.386	0.036 *
BFP × Age					0.290	0.010 **
SMM × Age					0.032	NS
R^2^ block 1 = 0.081	ΔR^2^ = 0.081					
R^2^ block 2 = 0.125	ΔR^2^ = 0.044					
R^2^ block 3 = 0.202	ΔR^2^ = 0.077					

BFP: body fat percentage; SMM: skeletal muscle mass. ΔR^2^: R-square change. * *p* < 0.05; ** *p* < 0.01. NS: not significant.

## Data Availability

The datasets used during the current study are available from the corresponding author on reasonable request.

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
