# Peer review of "The Effect of Body Composition on Gait Variability Varies with Age: Interaction by Hierarchical Moderated Regression Analysis"

_ijerph, 2022, doi:10.3390/ijerph19031171_

Round 1

Reviewer 1 Report

Introduction: Throughout the introduction, human data about gait and other variables of interest are given, without specifying men or women. However, when the objective of the study and the hypotheses are specified, they refer only to men. It is not clear why the study is done only with men or if gender difference is a relevant variable. Further explanation is needed.

Material and Methods: 

Figure 1:

  • Regarding the allocation, it would be necessary to know the age ranges (cutoff point) to consider young men group and older men group.
  • Why were there 3 people who did not receive allocated intervention?
  • Data for allocated interventions are not clear. For example, in the young group, there are 42 subjects in total, of which 20 receive allocated intervention and 2 do not receive it, but 40 are analyzed. Where are the other 20 who are missing? Exactly the same for the older group.

Results:

What is the purpose of the Independent sample t-test and Mann – Whitney test? Significant differences appear in age between the two groups, something to be expected since it is the variable that determines whether you are from one group or another, but also in other variables such as height, weight or BFP, does this have an impact on subsequent statistical analysis or on the overall results?  On the other hand, there are no differences in variables that have to do with gait. What do they contribute or how do they affect these results? I think a brief comment would be useful.

About the Hierarchical moderated regression analysis. The writing of the results is confusing and difficult to read. For this reason, it is difficult to follow the logic that leads from results to discussion and conclusions.

Discussion 

It focuses on the variables stride length and stride length CV, but overlooks that there are no differences in the other variables such as stride time CV. It seems like the text explains the results where differences are found, but not those where there are no differences.

Conclusions. The conclusions are written in such a way that stride length CV seems to be the only relevant variable for gait stability, but other variables such as gait speed or stride time have been analyzed. 

Reviewer 2 Report

Dear Authors, After performing an analysis of the paper submitted to the journal, I attach beneath the following comments. The Abstract is well structured in a fluent manner and according to the PICO criteria. In the main text however, I have found a few inconveniences. The authors do not define inclusion and exclusion criteria, or give adequate details of patients who became participants of the study. The rationale for the criteria for selection is not clearly and fully explained. Furthermore Authors do not specify the study design, they mention in the flowchart the randomization process, however none of the paragraph contains the direct message about the character of the presented clinical trial. That should be revised along with the step by step randomization sequence Also the discussion sections poorly referenced with the literature conducted on the similar topics, rationale and settings. Lastly, the limitations and strengths are not included in the study. The overall impression of the study is good. After introduction of the abovementioned minor revisions, it has a great opportunity to contribute significantly in academic merit.

Author Response

Dear Reviewer 2,

Manuscript ID: ijerph-1514455

Type: Article

Title: The Effect of Body Composition on Gait Variability Varies with Age: Interaction by Hierarchical Moderated Regression Analysis

Review Report Form

Open Review

(x)

I would not like to sign my review report

(  )

I would like to sign my review report

English language and style

(  )

Extensive editing of English language and style required

(  )

Moderate English changes required

(  )

English language and style are fine/minor spell check required

(x)

I don't feel qualified to judge about the English language and style

Yes

Can be improved

Must be improved

Not applicable

Does the introduction provide sufficient background and include all relevant references?

(x)

(  )

(  )

(  )

Is the research design appropriate?

(  )

(  )

(x)

(  )

Are the methods adequately described?

(  )

(  )

(x)

(  )

Are the results clearly presented?

(x)

(  )

(  )

(  )

Are the conclusions supported by the results?

(x)

(  )

(  )

(  )

Comments and Suggestions for Authors

Dear Authors, After performing an analysis of the paper submitted to the journal, I attach beneath the following comments. The Abstract is well structured in a fluent manner and according to the PICO criteria. In the main text however, I have found a few inconveniences. The authors do not define inclusion and exclusion criteria, or give adequate details of patients who became participants of the study. The rationale for the criteria for selection is not clearly and fully explained. Furthermore Authors do not specify the study design, they mention in the flowchart the randomization process, however none of the paragraph contains the direct message about the character of the presented clinical trial. That should be revised along with the step by step randomization sequence Also the discussion sections poorly referenced with the literature conducted on the similar topics, rationale and settings. Lastly, the limitations and strengths are not included in the study. The overall impression of the study is good. After introduction of the abovementioned minor revisions, it has a great opportunity to contribute significantly in academic merit.

Dear Reviewer 2, we sincerely appreciate your insightful comments. We have answered each of your points below.

  1. The authors do not define inclusion and exclusion criteria, or give adequate details of patients who became participants of the study. The rationale for the criteria for selection is not clearly and fully explained.

Author response: Thank you for your review. We improved inclusion and exclusion criteria, as well as study participant information (recruitment and enrollment, age criteria, sampling, adverse events, etc.). The revised content is as follows. (line 87)

“Participants were selected from a pool of healthy male adults living in the community. The recruitment method used was a community notice. Participants were chosen based on their ability to walk independently and their willingness to engage in physical activity. Those with a neuromuscular disease who had an artificial joint or metal device inserted were excluded. A random sampling method was used to select participants. Young men ranged in age from 20 to 30 years [30], while older men ranged in age from 55 to 85 years [31-35]. Due to time constraints, two young men who did not receive their assigned intervention were dropped out. One older man was unable to participate due to back pain. In addition, one elderly man dropped out of the experiment due to dizziness while walking, but no side effects were reported by any of the subjects after the study was completed.”

[30]         Han, S.H.; Kim, C.O.; Kim, K.J.; Jeon, J.; Chang, H.; Kim, E.S.; Park, H. Quantitative analysis of the bilateral coordination and gait asymmetry using inertial measurement unit-based gait analysis. PloS one 2019, 14, e0222913.

[31]         Van Ancum, J.M.; Jonkman, N.H.; van Schoor, N.M.; Tressel, E.; Meskers, C.G.; Pijnappels, M.; Maier, A.B. Predictors of metabolic syndrome in community-dwelling older adults. PLoS One 2018, 13, e0206424.

[32]         Stathokostas, L.; McDonald, M.W.; Little, R.; Paterson, D.H. Flexibility of older adults aged 55–86 years and the influence of physical activity. Journal of aging research, 2013.

[33]         Won, J.; Alfini, A.J.; Weiss, L.R.; Michelson, C.S.; Callow, D.D.; Ranadive, S.M.; Gentili, R.J.; Smith, J.C. Semantic memory activation after acute exercise in healthy older adults. Journal of the International Neuropsychological Society 2019, 25, 557-568.

[34]         Stumme, J.; Jockwitz, C.; Hoffstaedter, F.; Amunts, K.; Caspers, S. Functional network reorganization in older adults: Graph-theoretical analyses of age, cognition and sex. NeuroImage 2020, 214, 116756.

[35]         de Zubicaray, G.I.; Rose, S.E.; McMahon, K.L. The structure and connectivity of semantic memory in the healthy older adult brain. Neuroimage 2011, 54, 1488-1494.

  1. Furthermore Authors do not specify the study design, they mention in the flowchart the randomization process, however none of the paragraph contains the direct message about the character of the presented clinical trial. That should be revised along with the step by step randomization sequence

Author response: We elaborated on the precise causes for unassigned or terminated interventions during study design in the Materials and Methods section. The following are the newly added contents. If more instructions are received, we will quickly review them. (line 85 and 93)

“This study was designed as a cross-sectional study.”

 “Due to time constraints, two young men who did not receive their assigned intervention were dropped out. One older man was unable to participate due to back pain. In addition, one elderly man dropped out of the experiment due to dizziness while walking.”

  1. The discussion sections poorly referenced with the literature conducted on the similar topics, rationale and settings.

Author response: Thank you for your feedback. Items that were important in previous studies have been added (i.e. stride time and stride time CV). The following are the newly added contents. If more instructions are received, we will quickly review them. (line 255)

“Finally, this study found that body composition had no relationship with stride time or stride time CV. That is, increasing BFP decreased stride length, a spatial parameter of both young and old men, and increased stride length CV in older men, but it had no effect on the temporal parameters, stride time and stride time CV. According to the findings of previous studies, regardless of age, the stride parameter has a more significant correlation with the obesity factor in the spatial variable than in the temporal variable [11,12,15,40-42].

  1. Lastly, the limitations and strengths are not included in the study.

Author response: Thank you for your review. We rewrote the study's limitation and added a new one. Following is the written content. (line 265)

“This study has strengths and limitations. We consider it a strength that gait was assessed during 6 min natural walking not short length walking in the lab which could be close to real walking. However, we did not measure angular kinematic and kinetic data, so there were limitations to interpreting the results from multi-joint mechanisms during gait. The subjects in this study were all males. Gender was not taken into account. It was possible to reduce the gender deviation, but there could be differences in the results by gender. Thus there is a limit to generalizing the conclusion.”

Thank you for your valuable time.

Sincerely,

Round 2

Reviewer 1 Report

Thank you for the hard work you have done to clarify and answer the questions I had while reading and reviewing your article. I think you have answered all the questions I had and I now have a better understanding of both the purpose of the study and the results. 
